

# How well does a convection-permitting climate model represent the reverse orographic effect of extreme hourly precipitation?

Eleonora Dallan[1], Francesco Marra[2], Giorgia Fosser[3], Marco Marani[4], Giuseppe Formetta[5], Christoph Schär[6], Marco Borga[1]

[1] Department of Land Environment Agriculture and Forestry, University of Padova, Padova, Italy
[2] National Research Council of Italy - Institute of Atmospheric Sciences and Climate (CNR-ISAC), Bologna, Italy
[3] University School for Advanced Studies - IUSS Pavia, Pavia, Italy
[4] Department of Civil, Environmental and Architectural Engineering, University of Padova, Padova, Italy
[5] Department of Civil, Environmental and Mechanical Engineering, University of Trento, Trento, Italy
[6] Institute for Atmospheric and Climate Science, ETH Zürich, Zürich, Switzerland

*Correspondence to*: Eleonora Dallan (eleonora.dallan@unipd.it)

**Abstract.** Estimating future short-duration extreme precipitation in mountainous regions is fundamental for risk management. High-resolution convection-permitting models (CPMs) represent the state-of-the-art for these projections as they resolve convective processes key to short-duration extremes. Recent studies reported a decrease in the intensity of extreme hourly precipitation with elevation. This "reverse orographic effect" could be related to processes which are sub-grid even for CPMs. It is thus crucial to understand to what extent CPMs can reproduce this effect. Due to the computational demands, however, CPM simulations are still too short for analysing extremes using conventional methods. We introduce the use of a non-asymptotic statistical approach (Simplified Metastatistical Extreme Value, SMEV) for the analysis of extremes from short time slices such as the ones of CPM simulations. We analyse an ERA-Interim-driven COSMO-crCLM simulation (2000-2009, 2.2 km resolution) and we use hourly precipitation from 174 rain gauges in an orographically-complex area in Northeastern Italy as a benchmark. We investigate the ability of the model to simulate the orographic effect on short-duration precipitation extremes as compared to observational data. We focus on extremes as high as the 20-year return levels. While an overall good agreement is reported at daily and hourly duration, the CPM tends to increasingly overestimate hourly extremes with increasing elevation implying that the reverse orographic effect is not fully captured. These findings suggest that CPM bias correction approaches should account for orography. SMEV capability of estimating reliable rare extremes from short periods promises further application on short time-slice CPM projections, and model ensembles.



## 1. Introduction

Short-duration extreme precipitation in orographically complex areas is highly variable in space and time and may be the
trigger of numerous hydro-geological hazards, such as flash floods, debris flows, and landslides (e.g. Borga et al., 2014; Stoffel et al., 2016; Savi et al., 2021). Understanding the impact of orography on the probability distribution of extreme precipitation at short (i.e., ~hourly) temporal scales, as well as on extreme-rainfall causative processes, is critical for managing risk from rainfall-triggered natural hazards (e.g. Katz et al., 2002; Francipane et al., 2021). The enhanced convective activity and the changes in the dynamics of precipitation processes expected under foreseeable climate change
scenarios further strengthen the theoretical and practical interest in the relation between orography and extreme precipitation (e.g. Yan et al., 2021; IPCC 2019; Napoli et al., 2019).

Until recently, the robust estimation of future extreme precipitation for risk management strategies in regions with complex orography was severely limited due to the large resolution gap between Regional Climate Models (RCMs, resolutions of a few tens of km) and rainfall-triggered natural hazards (~hourly, few km). Some studies showed a high spatial correlation of
the 3- and 24-hour precipitation return levels estimated from RCMs at 12 km spatial resolution with those estimated from observational products. However, local deviations in complex-orography regions are evident (i.e. Berg et al., 2019; Poschlod et al., 2021) and point to the need of high-resolution modelling to improve the estimates of short-duration extremes in these areas (Poschlod et al., 2021).

With continuous advances in computing power, km-scale runs of regional climate models, i.e. Convection-Permitting
Models (CPM), become more common. In CMPs sub-grid parameterizations of atmospheric deep convection become unnecessary, thereby removing a major source of uncertainty and error in standard RCMs (Prein et al. 2015; Schär et al. 2020). Thanks to their ability to resolve convective systems and to better represent local processes, CPMs potentially provide more realistic representations of sub-daily precipitation statistics and extremes (Prein et al. 2015, Berthou et al. 2020). This leads to a greater confidence in CPM-based projections, compared to coarser resolution models (Ban et al. 2014, Kendon et
al. 2017, Fosser et al. 2020), even though their possible overestimation of extreme events is a topic of ongoing research (Kendon et al. 2021). In areas with a complex terrain, the possibility of explicitly resolving convection and a more detailed representation of orography and surface properties are crucial elements for correctly capturing the initiation and development of convection (Adinolfi et al. 2020, Hohenegger at al. 2008). For example, a coarse description of orography can lead to biases in the local precipitation pattern and intensity, due to the incorrect representation of the flow over mountains ridges
and of areas of atmospheric convergence triggering convection (Knist et al. 2020, Fosser et al. 2015). Over the Alps, but also elsewhere, CPMs tend to generate more precipitation at higher elevations than in reality, thus reducing the bias with respect to observations compared to RCMs (Lind et al. 2016, Reder et al. 2020). Ban et al. (2020) compared a CPMs ensemble and a RCMs ensemble in their representation of heavy daily and hourly rainfall over the greater Alpine region, and found that the CPMs improvements are more evident in summer, when convection plays a major role. Recent studies showed that it is

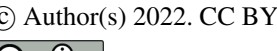



possible to improve the estimation of precipitation return levels in orographically complex regions using CPMs (Poschlod et al. 2021; Poschlod 2021). Therefore, the improved representation of extreme short-duration precipitation over complex orography is a key added value of CPMs, especially for the possibility to develop effective adaptation measures for rainfall-driven hazards and thus avoid severe impacts on society.

Mountain areas exhibit highly variable precipitation patterns, due to the interaction of atmospheric large-scale air motions
with complex local orographic features (e.g. Johnson and Hanson, 1995). Along the windward slope of the mountains the condensation of water vapor and the formation of clouds are enhanced by the orographic lifting of air masses. Conversely, precipitation tends to be reduced on the leeward side, where air descends after having released the moisture on the windward side and condensation is inhibited. The net effect consists of an increased precipitation amount at higher elevations, the so-called "orographic enhancement" of precipitation (e.g. Roe, 2005; Houze, 2012; Isotta et al., 2015; Avanzi e al., 2021),
observed by climatological analysis worldwide (e.g. Frei and Schär, 1998; Malby et al., 2007; Harris et al., 1996). Several factors influence this orographic enhancement, including static atmospheric or aerosol conditions, local terrain slope and shadowing effects (e.g. Napoli et al., 2019). However, a simple precipitation–height relation is difficult to establish, because the topographic signal is also associated with slope and shielding. In addition, the precipitation increase is robust only for low and intermediate topographic heights. In the Alps, maximum annual mean precipitation is typically in the height range
of 800–1200 m (Frei and Schär, 1998), and above this altitude precipitation may again decrease with height. While the orographic enhancement is also observed for relatively long-duration precipitation extremes (few hours or more), an opposite effect has been reported for short-duration extremes (hourly and sub-hourly), the so-called "reverse orographic effect" (Avanzi et al., 2015). The reverse orographic effect characterises regions where sub-daily extremes are linked with convective processes (Formetta et al., 2022; Marra et al., 2022a) and impacts both annual maxima (Allamano et al., 2009;
Avanzi et al., 2015; Mazzoglio et al., 2022) and extreme return levels of interest for risk management applications (Rossi et al., 2020; Formetta et al., 2022; Marra et al., 2022a). Overall, these studies suggest that orography influences precipitation extremes, and thus associated hazards, differently at different time scales. Therefore, for a reliable estimation of extreme precipitation across scales, an ideal model should capture both these orographic effects.

Marra et al. (2021) suggested that orographically-induced turbulence could cause a weakening of the updrafts, with a
consequent weakening of the peak intensities of the convective cells and a redistribution of the moisture over the surrounding areas. As a result, the typical convective cells in orographic areas are weaker in intensity and smoother in spatial structure compared to nearby flat areas. Additionally, the complex three-dimensional structure of heavy rotating thunderstorms can be disrupted by sharp valleys and ridges, and the supply of warm moist air to drive these storms is smaller in regions of complex topography. However, these are sub-grid phenomena even for CPMs, raising the important question:
to what extent can CPMs capture the reverse orographic effect on extreme rainfall of short duration?



While CPMs have a spatiotemporal resolution in line with the requirement of the hazard models, existing CPM simulations are limited to relatively short time periods (10–20 years) due to the high computational costs. This prevents the use of conventional extreme value approaches for quantifying the probability of occurrence of extreme return levels (i.e. Katz et al., 2002). Poschlod (2021) evaluated four statistical approaches and their uncertainty to calculate 10-yr and 100-yr return levels

at daily duration based on a 30-yr-long 1.5-km-resolution climate model. Their findings suggested that classic methods based on extreme value theory, such as the fit of Generalized Extreme Value and Generalized Pareto distributions respectively to annual maxima and peaks over threshold, can be prone to large uncertainties, especially for return periods longer than the available record. These limitations may be at least partially overcome using a recent extreme-value analysis method, which makes use of all available data, rather than just yearly maxima or a few values above a high threshold (Marani and

Ignaccolo, 2015).

Indeed, alternative approaches were recently proposed for deriving accurate frequency analyses from relatively short data records, opening the possibility of exploring extreme-value properties in short CPM time slices. These methods include the Metastatistical Extreme Value Distribution (MEVD; Marani and Ignaccolo, 2015; Zorzetto et al., 2016) and its possible simplification, the Simplified MEV (SMEV; Marra et al. 2019, 2020). These approaches are based on the statistical analysis

of the so-called ordinary events (see details in Marani and Ignaccolo, 2015), which are all the independent events that share the statistical properties of extremes: once the upper tail of the ordinary events is known, it is possible to derive an extreme value distribution by explicitly considering their yearly occurrence frequency. The method has been successfully applied to point and spatial rainfall, as well as to a variety of geophysical processes, showing improvements in high-quantile estimation uncertainty with respect to traditional approaches (Caruso and Marani, 2022; Hosseini et al., 2020; Miniussi and Marani,

2020; Zorzetto et al., 2016). Owing to a decreased number of parameters to be estimated (Marra et al., 2019), the SMEV approach may be used to derive more accurate high quantile estimates than the full MEVD model, due to the possibility to better isolate the tail of the ordinary events distribution (see below) (Poschlod 2021; Wang et al., 2020; Miniussi and Marra, 2021; Vidrio-Sahagún and He, 2022). Interestingly, due to their effective use of available information, these methods are also suited to examine the altitudinal variations of extremes (Marra et al. 2021, 2022a; Formetta et al., 2022; Amponsah et

al., 2022). In fact, by directly exploiting the available short-duration records at high elevations, they do not require regionalizations (e.g., Buishand, 1991) or duration-scaling approaches, which would inevitably smooth existing orographic impacts.

In this paper we use a SMEV approach to examine the ability of CPM runs to realistically represent observed extreme value distributions of hourly precipitation in an orographically complex region like the north-eastern Italian Alps area. Moreover,

we investigate the relationship of orography with observed and simulated extreme return levels, with a special focus on the reverse orographic effect at the hourly duration. We propose a physically-based interpretation of the resulting differences in the discussion.



## 2. Study area and data

The study area is located in northeast Italy and consists of a north-south transect that ranges from the Italian Alps to the Po river and the Adriatic Sea. The area (around 32000 km$^2$) includes the Veneto region and the provinces of Bolzano and Trento, and covers a range of altitudes between -5 m and 3990 m a.s.l. (Figure 1a). The area is particularly interesting for its orographic complexity, which determines a high climatic heterogeneity on a wide range of spatial scales. The south-eastern portion of the region is in close proximity to the Adriatic Sea, so that possible effects associated with the sea-land contrast and its representation in CPM runs can be observed. However, this part of the region is rather flat and will not be used in the derivation of the orographic relations (see Section 3). The north-western portion of the region receives relatively low amounts of precipitation (about 500 mm yr$^{-1}$, on average), due to the orographic shielding offered by the surrounding mountains. Larger amounts are typically observed in the central part of the domain, the so-called Prealps, which represent the first orographic obstacle to the dominant precipitation systems reaching the area, and cause a strong orographic enhancement (up to 2300-2500 mm yr$^{-1}$; e.g., Isotta et al., 2014). In the south-eastern part of the region, from the coastal zone to the lowlands and Prealps, the mean annual precipitation is about 800 mm yr$^{-1}$, and increases towards the Prealps. Extreme precipitation shows specific spatial patterns, which are consistent with the orographic characteristics of the region and strongly dependent on the temporal scale. In particular, Formetta et al. (2022) describe two distinct modes of orographic relationship: an orographic enhancement for durations longer than ~8 h and a reverse orographic effect for hourly and sub-hourly durations, which consist of a reduction in the total amount of water released by convective cells and of a weakening of their peak intensity.

### 2.1 Rain gauge data

As a benchmark in this study, we used continuous quality-controlled rainfall observations with 5 minutes temporal resolution and 0.2 mm data quantization collected at 174 heated rain gauges (density ~1/180 km$^2$, Figure 1a). We considered only rain gauges with at least 9 valid years during the period 2000-2009, where a year is defined as valid when less than 10% of the data are missing or flagged as low quality. The total record length of the selected stations ranges from a minimum of 14 to a maximum of 37 years. The rain gauges cover elevations in the range -3 ÷ 2235 m a.s.l. (Figure 1b). Prior to the analyses, the data were aggregated at a 1-hour temporal resolution to match the resolution of the CPM output.

### 2.2 Convection-permitting model rainfall data

The CPM simulation used in the study was run by ETH Zurich with COSMO-crCLIM. It covers the greater Alpine region defined under the Coordinated Regional Climate Downscaling Experiment (CORDEX) Flagship Pilot Study on Convective Phenomena over Europe and the Mediterranean (FPS-Convection; Coppola et al. 2020). COSMO-crCLIM is the climate version, running on GPU, of the state-of-the-art weather prediction COSMO (Consortium for Small Scale Modeling) non-





hydrostatic, limited-area model (Rockel et al., 2008). More details on the used physical parameterisations can be found in Leutwyler et al. (2016). The simulation at 2.2 km resolution, covering the period 2000-2009, is nested within a 12 km

European RCM, in turn driven by ERA Interim (Dee et al. 2011). The simulation has been evaluated against several observational datasets and additional CPMs and RCMs from FPS-Convection ensembles by Ban et al. (2021). In our study, CPM hourly precipitation data have been extracted at the nearest grid point to each rain gauge, to obtain the "Station-Collocated" CPM time series (SC_CPM in the following). Figure 1b, c shows the elevation difference between the rain gauge and the related station-collocated grid point. We then also analysed all the ~6500 GRid CPM points in the study area

(GR_CPM).

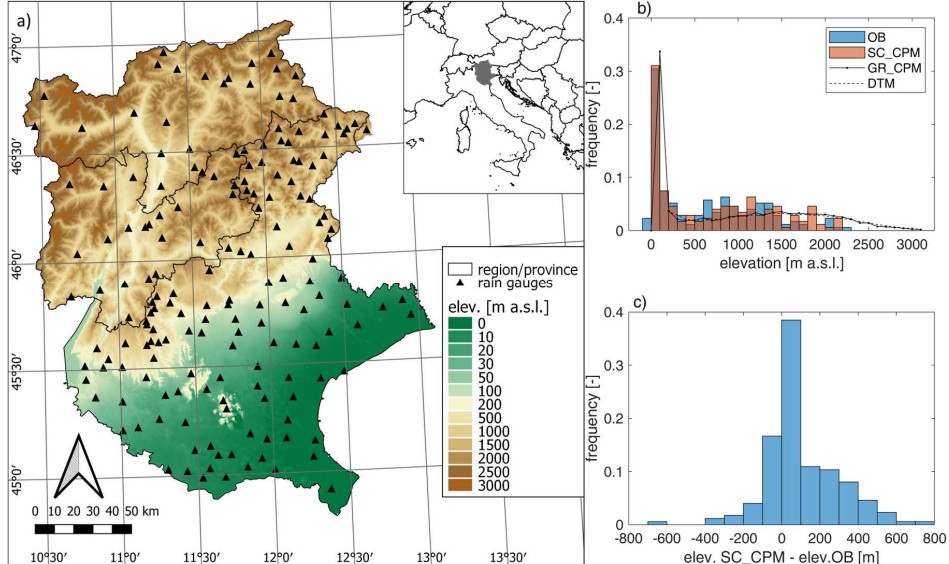

**Figure 1. Study area and data. a) Orography of the study area and location of the rain gauges; b) frequency distribution of the elevation for: the 174 rain gauges (observations, OB), the station-collocated CPM (SC_CPM), all the CPM grid points (GR_CPM), and the Digital Terrain Model (DTM) for the study area; c) distribution of the elevation differences between station-collocated CPM and observations.**

## 3. Methods

Observed (OB) and simulated (CPM) precipitation time series are analysed and compared focusing on: i) annual maxima (AM), defined as the largest values observed in each calendar year, ii) return levels estimated through a novel statistical method, SMEV and iii) SMEV distribution parameters. Specific attention is paid to the orographic impact on the above





quantities, which is examined via linear relations with elevation. We focus on the 1-hour temporal scale, the finest temporal
resolution for which precipitation is provided in CPM runs, but we also explore CPM-generated extreme rainfall at the daily
scale, for which generally more observational data are available and orographic effects are well characterized. Both
observations and station-collocated CPM data are analysed over the 10-year common period 2000-2009. Analyses on full-
record observations are also carried out and presented in the supplementary material to further assess the robustness of the
results.

### 3.1 Statistical method

Non-asymptotic statistics were recently proposed as an alternative to extreme value theory for the estimation of extremes
corresponding to low yearly exceedance probabilities (e.g., Marani and Ignaccolo, 2015). These approaches are based on the
idea that extremes are samples from the so-called ordinary events, which are the independent realizations of the process of
interest. Since ordinary events are much larger in number than extremes, these approaches offer the advantage of using most
of the observational information, rather than one or a few large values from every year of observation as in the case of
extreme value theory (Zorzetto et al., 2016). The fundamental assumption behind these approaches is that a suitable
statistical model describing the ordinary events may be identified. When this is the case, the probability distribution of the
ordinary events can be used to construct the distribution of yearly maxima and to capture the probability of occurrence of
rare and potentially unprecedented extremes. We adopt here the Simplified Metastatistical Extreme Value (SMEV) approach
(Marra et al., 2019; 2020). Following Marani and Ignaccolo (2015), who use theoretical reasoning (Wilson and Toumi,
2005) to justify this choice, we adopt a Weibull distribution to model the "tail" of the ordinary events distribution. The latter
is defined by Marra et al. (2020) as the portion of the empirical ordinary events distribution that can be fully described by a
two-parameter Weibull distribution according to a proper test (see below). This choice of model is supported by recent
results on the study area (Formetta et al., 2022; Dallan et al., 2022). This means that the probability of observing extreme
intensities decreases as a stretched exponential, following the cumulative distribution function:

$$F(x; \lambda, \kappa) = 1 - e^{-\left(\frac{x}{\lambda}\right)^{\kappa}} \tag{1}$$

with scale parameter $\lambda$ and shape parameter $\kappa$. Once the tail of the ordinary events distribution $F$ is known, it is possible to
write an analytical approximation for the cumulative distribution function of the annual maxima:

$$\zeta(x; \lambda, \kappa, n) \simeq F(x)^n = \left[1 - e^{-\left(\frac{x}{\lambda}\right)^{\kappa}}\right]^n \tag{2}$$

where $n$ is the average number of ordinary events observed in a year. Marra et al. (2019) showed that the inter-annual
variability of the number of ordinary events per year can be neglected, especially when interested in rare extremes.





We note that this approach is a non-asymptotic formulation, as opposed to the classic alternative of the extreme value theorem, in which an asymptotic assumption on $n$ ($n \rightarrow \infty$) or, for the case of threshold exceedances on the threshold $\theta$ ($\theta \rightarrow$ 200 $\infty$ for the case of unbounded distributions) is required. The formulation explicitly separates the ordinary events intensity distribution ($F$) from their occurrence frequency ($n$), and thus provides grounds for improved interpretations of the relation between processes (and their changes) and extremes (e.g., Marra et al., 2021; Formetta et al., 2022; Dallan et al., 2022; Vidrio-Sahagún and He, 2022).

### 3.1.1 Evaluation of the SMEV assumptions and definition of the tails

It is possible to use a specific test to evaluate the robustness of our underlying assumption of Weibull tails. The test, described in detail in Marra et al. (2022b) checks whether the observed extremes (i.e., the annual maxima) are likely samples from the assumed distribution. While in principle the test can only reject the hypothesis, results based on synthetic data show that it is robust in separating Weibull tails from heavier tails, among the supported alternatives to the Weibull tails (Marra et al., 2022b). Results of this test indicate that in our study region the top 10% (for hourly durations) or 15% (for 24 hours 210 duration) of the ordinary events can be described using a Weibull tail. This is consistent with previous results in northern Europe (Miniussi and Marra, 2021) and is slightly smaller than what previously adopted in some subsets of the region (Formetta et al., 2022; Dallan et al., 2022).

### 3.1.2 Estimation of extreme return levels using SMEV

The SMEV statistical model is applied using the approach described in Marra et al. (2020), whose codes are freely available 215 (Marra, 2020): (i) storms are defined as consecutive wet periods separated by dry hiatuses (see more details in the next paragraph) of at least 24 hours; (ii) ordinary events of the duration of interest are computed as the maximal intensities observed within each storm using running windows of the duration of interest moved with 1 hour steps; (iii) parameters of the Weibull distribution are calculated by left-censoring the ordinary events below the above-mentioned thresholds and using a least-squares linear regression in Weibull transformed coordinates; (iv) return levels of interest are computed by inverting 220 eq. (2). Using this approach, the number of ordinary events is the same across all durations and matches the number of storms, as follows from point (i) and (ii) (for more details, see Marra et al., 2020).

### 3.1.3 Definition of wet hours

The rain gauges used in this study start recording rain above 0.2 mm, while the CPM model has continuous rainfall values above zero. In the climate modelling community, a wet hour is usually defined as an hour with precipitation above 0.1 mm h⁻ 225 ¹ (e.g. Ban et al. 2014, 2020; Meredith et al. 2020). We conducted a sensitivity analysis on the CPM data to investigate the impact of different thresholds for the definition of wet hours on the number of yearly events $n$ and of the return levels. We



explored thresholds between 0.01 and 0.5 mm h$^{-1}$. The results showed a small sensitivity of $n$ to the selected threshold (±5% change in hill/mountain zones, ±10% change in lowlands), and no appreciable change on the estimated return levels, as expected given the SMEV structure (see Figure S1 in the Supporting Information). A threshold of 0.1 mm h$^{-1}$ was then used

for the definition of a wet hour in CPM data in the rest of the analysis.

### 3.2 Assessment of CPM biases

From the analysis of each dataset (rain gauges, CPM), we derived the following quantities at each location and for 1 h and 24 h durations: i) annual maxima and their mean value, ii) return levels up to 100-yr return period, iii) average yearly number of ordinary events $n$ (which is the same across all durations), iv) scale $\lambda$ and shape $\kappa$ parameters of the Weibull distribution

describing the tail of the ordinary events. For each quantity $X$, the multiplicative bias $B_X$ between observation and station-collocated CPM is computed as the ratio between the variable value $X_{CPM}$ obtained from CPM and the variable value $X_{OB}$ obtained from the collocated observations:

$$B_X = \frac{X_{CPM}}{X_{OB}} \tag{3}$$

It is here pointed out that the comparison between a point value (observation) and an areal value (single CPM grid value) is

made directly, as the correlation length of extreme rainfall at hourly duration is typically greater than the grid resolution of our CPM (e.g., Villarini et al. 2008).

### 3.3 Quantification of the orographic effect

The orographic effect on short-duration extreme rainfall is explored by looking at the relationship with elevation of different quantities obtained for 1 h duration: Annual Maxima (AM; also for 24h duration), return levels, distribution parameters and

average number of yearly events. The relations are approximated with a linear model. Linear regression slopes with elevation are computed for each quantity for both observations and station-collocated CPM. Given the wide extent of the floodplains in the examined region and the proximity of some of these areas to the sea, the results for locations below 100 m a.s.l. are expected to include a variety of distinct behaviours which clearly do not depend on orographic forcing. Regression slopes are thus computed only by considering locations with elevation exceeding 100 m a.s.l. and expressed in the following

as percentage of the median value per km of elevation. The results for all grid points of the CPM in the study area (GR_CPM) are also considered to evaluate if the SC_CPM is a representative sample of the climate model results.

### 3.3 Uncertainty and statistical significance

Uncertainty associated with the SMEV estimates is quantified using a 1000-iteration bootstrap resampling procedure with replacement on the years (Efron and Tibshirani, 1993; Overeem et al., 2008), for both observed and simulated results. This



bootstrap approach is also used to evaluate the statistical significance of the bias in the model simulations and of the orographic relationships with respect to the stochastic uncertainties related to the available data sample. Specifically, 1000 bootstrap surrogates were created by randomly selecting 10 years between 2000 and 2009 with replacement, for both observations and station-collocated CPM. This implies that in each bootstrap sample, the same sequence of years is used for all the stations and datasets. The annual maxima and the SMEV distribution parameters, number of events, return levels, and

slopes of their relation with elevation are then computed for each bootstrap sample. For each of these quantities, the distribution of the 1000 differences between OB and SC_CPM is analysed to assess whether the hypothesis of having no difference between CPM and observations could be rejected. The null hypothesis of no difference is rejected at the 5% level when the 2.5[th] percentile of the distribution of differences is greater than zero or the 97.5[th] percentile is less than zero (e.g. Kendon et al., 2012).

## 4. Results

The following sections first present the comparison between observed and simulated annual maxima (intensity, bias, relation with elevation) and then focus on the SMEV analysis for the 1h duration return levels.

### 4.1 The reverse orographic effect on observed mean hourly annual maxima

The observed mean annual maximum intensity at 1 h duration are shown in Figure 2. A spatial organisation of the rain rates can be noticed (Figure 2a). Indeed, higher values, even >35 mm h$^{-1}$, are observed in the south-eastern part of the study area,

mostly corresponding to floodplains and coastal area, while lower values (even <15 mm h$^{-1}$) are observed in the northern and north-western parts, corresponding to mountainous areas in the dry heart of the Alps. Figure 2b reports the relationship of the 1 h mean AM with elevation. The observed reverse orographic effect clearly emerges, with an average decrease of the mean AM hourly precipitation of more than 30% km$^{-1}$ (computed using the rain gauges above 100 m a.s.l.), which corresponds to a

decrease of about 7 mm h$^{-1}$ km$^{-1}$.





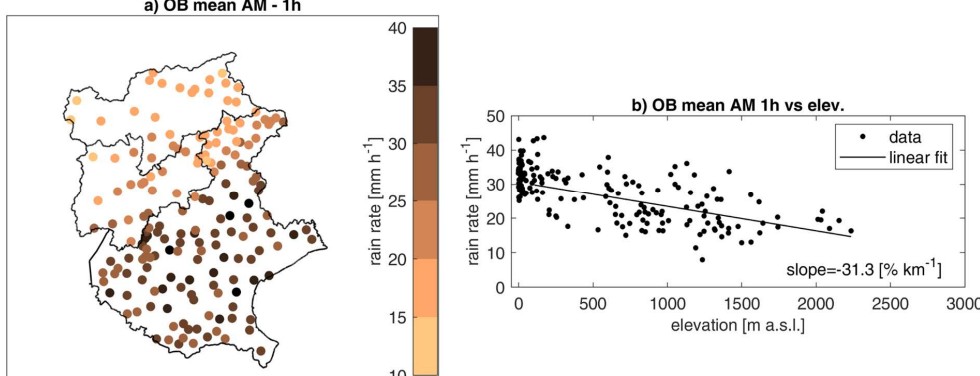

**Figure 2. Observed AM at 1 h duration: a) map with the mean AM; b) relationship of the mean AM with elevation, slope for the linear regression is expressed as a percent of the median value and is calculated for the stations above 100 m a.s.l. (solid line)**

**4.2 Bias assessment and reverse orographic effect on simulated annual maxima**

The comparison between observed and simulated mean annual maxima at 1 h and 24 h durations is shown in Figure 3. In panels (a) and (c), the scatter plots describe how SC_CPM and OB differ in the representation of the AM rainfall rate. CPM represents OB AMs at the daily duration better than at the 1 h duration both in terms of central tendency (mean bias ~1 and ~1.1, respectively, indicating a prevalence of overestimation for the hourly durations) and variance. Especially for 1 h duration (Figure 3a, c) the CPM mostly overestimates the AMs at the high elevation locations, which are also characterised

by low observed intensity; in lowlands the observed values are higher, and tend to be underestimated by the CPM. The maps in panels (b) and (c) make this evident: for both durations, observed AMs tend to be underestimated in lowland and coastal zones, while they tend to be overestimated at high elevations. The overestimation is much stronger for 1 h than for 24 h and the biases are significant at the 5% level in ~40% and ~34% of stations, respectively.

The relationship between mean AM precipitation and elevation is displayed in Figure 4 for the 1 h (panels a, b) and the 24 h

duration (c, d). For hourly duration CPM rain rates are clearly underestimated in regions below <100 m a.s.l. and overestimated in regions above 1100 m a.s.l. Considering both the interquartile range and the whiskers in the boxplots in panel (b), one can notice the high variability among stations located at similar altitudes; this spread is substantially reduced in the CPMs, as CPM simulations are more uniform in their rain intensities especially over the mountains. In Figure 4a, linear regressions with elevations are reported. The slope for CPM is negative, indicating that the CPM can actually capture

a reverse orographic effect on mean 1 h AM intensity, although the strong decrease with elevation found in the observations (-31% km$^{-1}$) is not fully captured by SC_CPM (-9% km$^{-1}$). The slopes are significantly different at the 5% level. A better agreement is found at 24 h duration: observed and CPM intensities are similarly distributed in the explored range of





elevations, and have no evident relation with elevation (panel c). The boxplots in panel d, which compare daily intensity within the same elevation group, show a good agreement between observations, SC_CPM, and GR_CPM. For lowlands

(<100 m a.s.l.) and for high mountains (>1100 m a.s.l.) the CPM tends to respectively underestimate and overestimate with respect to the median OB rain rate, but the overlapping interquartiles indicate that the biases are generally within the spatial variability range of that elevation class. We can then observe that the results on sampling station-collocated CPM and those on the whole grid CPM are consistent in terms of regression slopes at 1 h, box plot medians and interquartiles across elevations and durations. This indicates the SC_CPM results are not affected by the sampling due to the location of the rain

gauges; they are a representative sample of the elevation characteristics of the study area

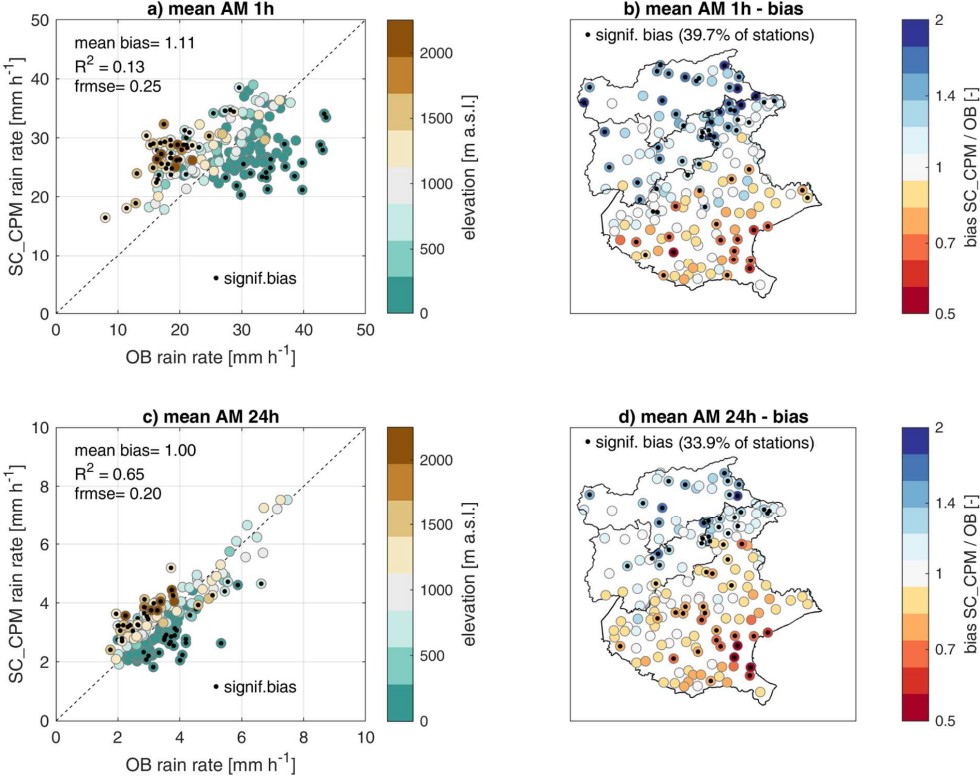

**Figure 3. Comparison of observed and simulated annual maxima at 1 h and 24 h durations. (a, c) Rainfall rate for average annual maxima for Station-Collocated CPM (SC_CPM) versus observed values (OB) at 1 h (a) and 24 h durations (c); the colour of the**

**dots indicates the elevation of the station; mean bias, coefficient of determination (R2), and fractional mean squared error (fmse)**





**are also shown. (b, d) Maps of SC_CPM/OB relative bias for the 1 h (b) and 24 h (d) mean AM. In all panels, significant differences at 5% level are indicated with a black dot and their proportion is reported as the percentage of significant cases on the total number of stations.**

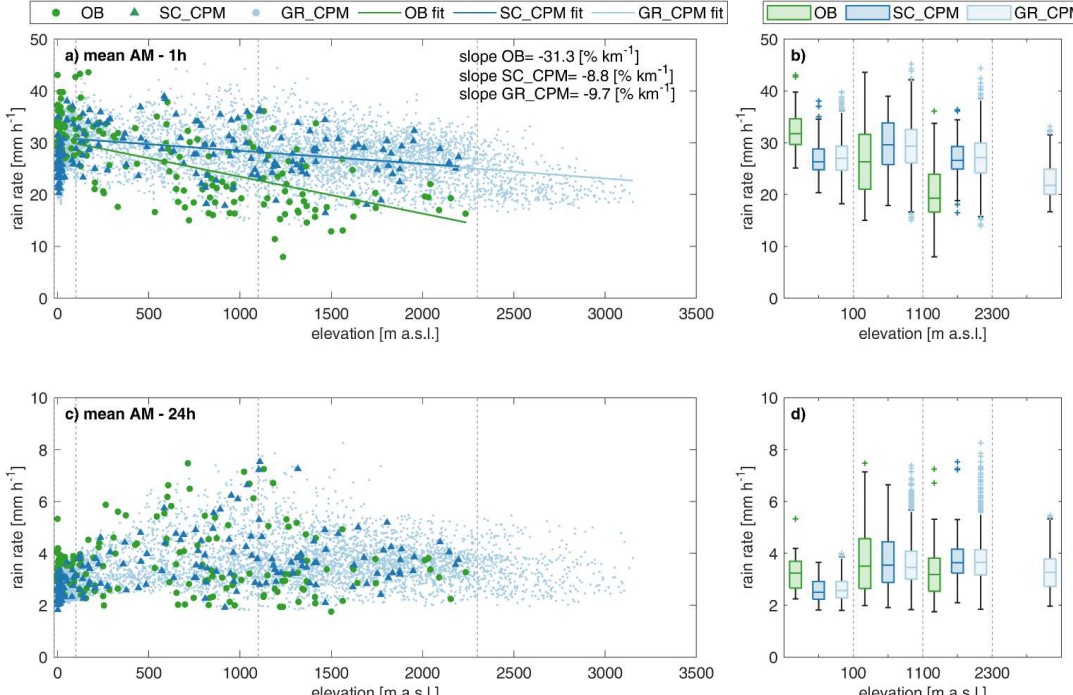

**Figure 4. Orographic effect on 1 h and 24 h annual maxima for observation (OB), station-collocated CPM (SC_CPM), all CPM grid points (GR_CPM). (a, c) Relationship of AM rain rate with elevation at 1 h and 24 h durations, respectively. In panel a, the linear regressions lines are shown as a solid line, are expressed as a percent of the median value and are calculated for the stations above 100 m a.s.l.; (b, d) Box plots of AM rain rate at 1 h and 24 h durations, respectively, for the three rainfall datasets and 4 elevation groups. Note that the considered elevation data is the one of each dataset (OB or CPM).**

## 4.3 Hourly return levels and relation with elevation

We estimate the return levels of hourly precipitation for several return periods. Results on bias assessment and relation with elevation are here reported for the 20 yr return levels as reference, but similar results are found for return periods up to 100 yr and reported in the following Discussion section.

Figure 5 shows the comparison between estimated 20 yr return level from observations and SC_CPM (panel a), and the magnitude of the relative bias at each location (panel b), while the spatial distribution of the rain intensity for the 1 h



duration 20 yr return level is reported in maps in Figure S2. As already observed for the AM, CPM overestimation is stronger at the low-intensity mountain locations while the underestimation is particularly evident in lowlands and coastal areas where higher intensities are observed (panel a). The significant biases (at about 30% of the locations) are found mainly in the proximity of the Adriatic Sea and in the northeastern portion of the mountainous domain, characterised by narrower
valleys than the western part (panel b).

The spatial pattern in the 20yr return level bias for 1 h duration shown in Figure 5b is consistent with the one shown in Figure 3b for 1 h duration AM, and the slightly higher coefficient of determination ($R^2$ =0.13 for AM, $R^2$ =0.19 for the 20yr return level) indicates the statistical model is robust and has lower random errors than the stochastic sampling of AM. The higher fractional mean squared error (frmse=0.25 for AM, frmse=0.30 for for the 20 yr return level) for the 20yr return level
indicates a wider range in the bias magnitude: from 0.53÷2.08 for 1 h mean AM to 0.45÷2.63 for 1 h 20 yr return level.

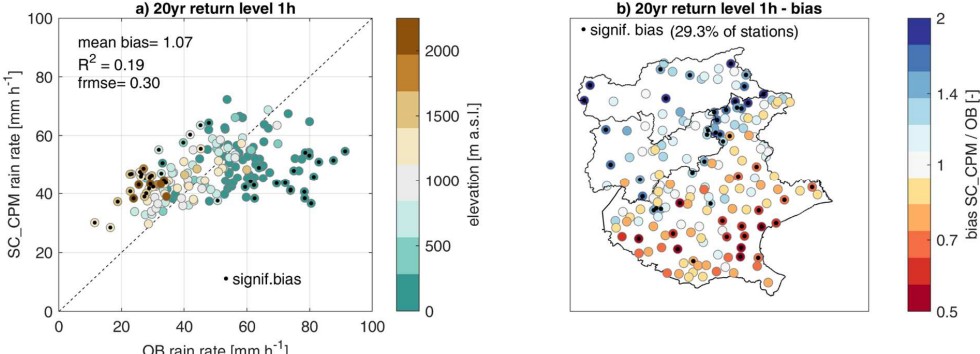

**Figure 5. Bias assessment of 20 yr return level at 1 h duration. a) Rainfall rate for 20 yr return level, 1 h duration, for Station-Collocated CPM (SC_CPM) versus observed values OB; the colour of point indicates the elevation of the station; mean bias, coefficient of determination ($R^2$), and fractional mean squared error (frmse) are also shown. b) Maps of SC_CPM/OB relative bias for the 1 h duration 20 yr return level. In all panels, significant differences at 5% level are indicated with a black dot and their proportion is reported as the percentage of significant cases on the total number of stations.**

The 20 yr return level at 1 h duration estimated from observation shows the reverse orographic effect, with a negative normalised slope of -36% km[-1] (Figure 6a), which is stronger than the one reported for the mean AM. This is consistent with
the results from Marra et al. (2021), Marra et al. (2022a), and Formetta et al. (2022), which showed a decrease in tail heaviness with elevation at hourly durations. The reverse orographic effect on the hourly 20 yr return levels is weaker for the CPM (normalised slope is ~ -14% km[-1]) than for observations, and it is similar when considering all CPM grid points (normalised slope is ~ -12% km[-1]). The slopes test significantly different at the 5% level. The boxplots in Figure 6b show that the CPM tends to underestimate (overestimate) return levels at low (high) elevations. Compared to the analysis of AM,





the spread within each elevation category increases more in OB than in SC_CPM, highlighting the strong variability among
       stations. These results show that, when estimating short-duration high return levels relevant for risk management, the
       orographic effect is not negligible, and the CPM considered in our study does not fully capture it.

       It is worth noting that, despite only using 10 years of data, 20 yr return levels computed with the SMEV approach used here
       are subject to relatively small stochastic uncertainties (quantified here by means of the coefficient of variation of the 1000

bootstrap surrogates). Figure S3 reports the uncertainty in the 1 h duration 20 yr return levels, evaluated based on the 10 yr in
       the period 2000-2009. The median value of the uncertainty is 13%, only slightly smaller than the one found using a random
       sample of 10 years within the entire available rain gauge record (15%), and slightly larger than the 9% uncertainty computed
       when considering the whole observational period. The median uncertainty related to 1 h 20 yr return levels estimated from
       the CPM is 11%. Results on the full-record observations, reported in Figure S4, are quantitatively unchanged, with the

exception of low-elevation locations where the median estimated return level is similar but the spatial variability is reduced
       (see figure S4b). The consistency of the return level estimates obtained from the full record and from the 10 yr record, and
       the small increase in the associated uncertainty that, once its assumptions are verified, SMEV is a reliable statistical method
       for the analysis of extreme precipitation from short time slices.

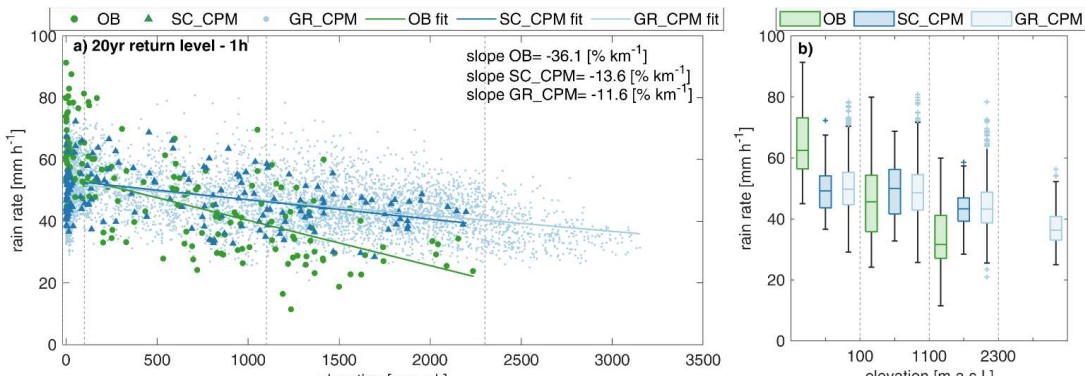


**Figure 6. Orographic effect on 1 h duration 20yr return levels. a) Relationship of the return levels with elevation for observation (OB), station-collocated CPM (SC_CPM), all grid points (GR_CPM). The linear regressions are shown as a solid line, are expressed as a percent of the median value and are calculated for the stations above 100 m a.s.l.; b) Box plots of the return levels for the three rainfall datasets and 4 elevation groups. Note that the considered elevation data is the one of each dataset (OB or**

**CPM).**



## 5. Discussion

### 5.1 Reverse orographic effect at different return periods

By exploiting the potential of SMEV in giving accurate return level estimates for high return periods, we analysed return periods up to 100 years, to investigate how the reverse orographic effect at 1 h duration is represented in both observations

and the CPM. Figure 7 shows the normalised slope of the linear regression between different return levels and elevation (computed for elevations >100 m a.s.l.) and the associated uncertainty quantified as the 95% confidence interval from 1000 bootstrap regressions. The slope for the mean AM is also reported for comparison. In line with Formetta et al (2022), the observed reverse orographic effect at 1h duration is consistent across the different return levels, with a higher negative slope at 100yr return time. The discrepancy between the slopes of observation and of station-collocated CPM is similar across the

different return levels (median differences range between $19 \div 23\%$) and these differences are all statistically significant at the 5% level. The slopes obtained from the analysis on the whole CPM grid are within the uncertainty range of the SC_CPM slopes. The consistency of the findings across the return periods, and the modest increase in uncertainty at the higher return period, show that SMEV allows reliable evaluation of the elevation dependencies of high return levels from a short CPM time slice.

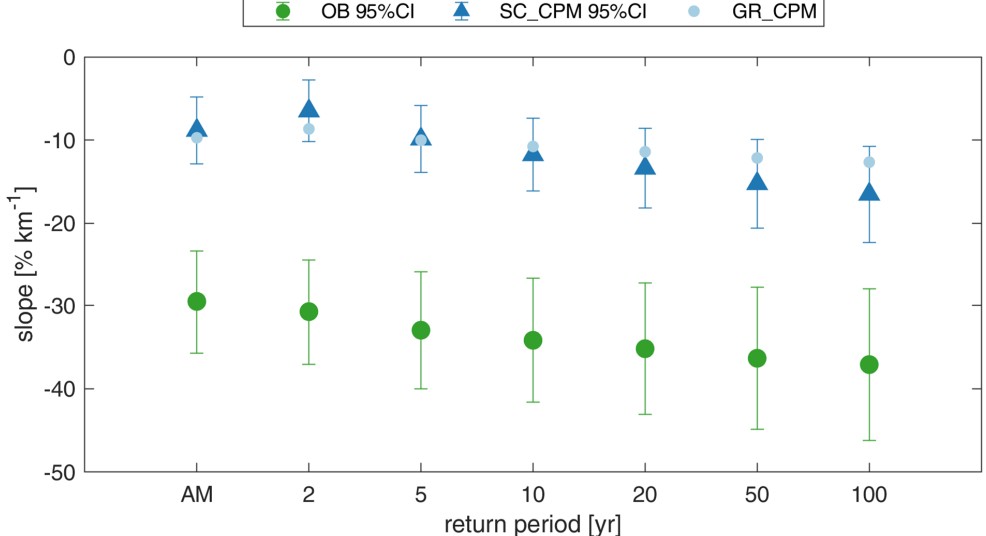


**Figure 7: Normalised slope of the relation with elevation for 1h duration annual maxima (AM) and return levels (2, 5, 10, 20, 50, 100 yr return period), for observations (green circle), station-collocated CPM (blue triangle), grid, CPM (light blue circle); all the slope differences result significant; error bars indicate 95% confidence interval from 1000 bootstrap regressions.**



## 5.2 Bias assessment on the distribution parameters

The statistical method based on the separation of storm intensity and occurrence frequency allows us to analyse the differences in the parameters of the ordinary events distribution. This, in turn, gives us insights into the mechanisms behind the biases found in CPM. In Figure 8, the biases in the scale and shape parameters at 1 h duration and in the number of events are shown in maps (panels a, b, c) and as boxplot for different elevation groups (panels d, e, f).

A distribution parameter $\lambda$ is called a "scale" parameter when $F(x; \lambda) = F(x/\lambda; 1)$. The scale parameter thus "scales" all the

intensities $x$ by the same factor; a higher (lower) scale implies proportionally higher (lower) return levels. In the study area, the CPM generally overestimates the scale parameter, with interquartile ranges of the bias exceeding 1 for all the elevation groups (Figure 8d). The overestimation of the scale parameter is larger in the high mountains (Figure 8a) where the median bias is close to 2 (in median, estimated return levels would be double than the observations - assuming no bias in the other parameters) and the boxplot whiskers are completely above 1 (last group in Figure 8d). Also in the coastal zone, the south-

eastern part of the domain, the scale is overestimated. Underestimation is present in the central part of the lowland area, and in the western mountain, but with only few significant cases. The biases on the scale are statistically significant at the 5% level in 42.5% of cases.

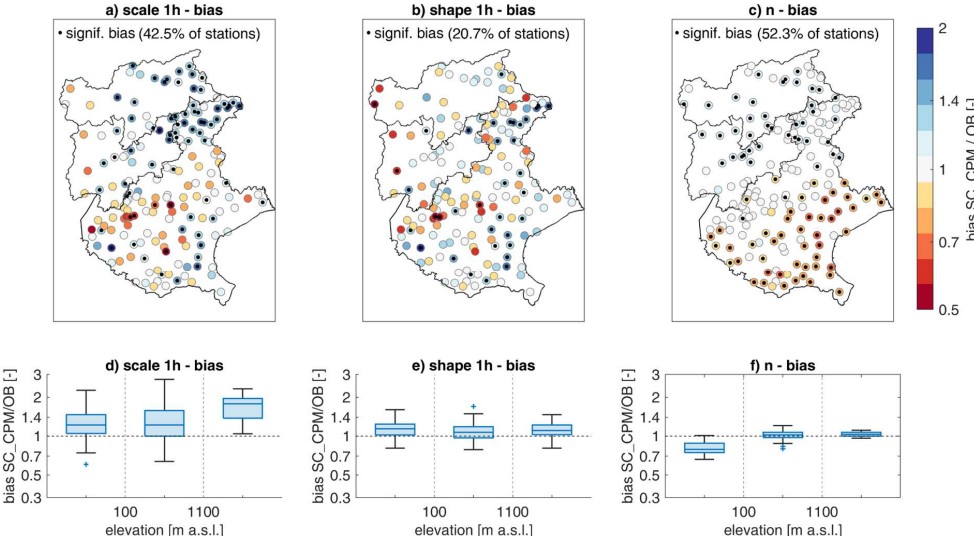

**Figure 8: Maps and boxplot of the bias in the estimated SMEV distribution parameters: scale (panel a, d), shape (panel b,e), n (panel c, f). In all panels, significant differences at 5% level are indicated with a black dot and their proportion is reported as the percentage of significant cases on the total number of stations.**



The shape parameter defines the heaviness of the Weibull distribution right tail: lower shape parameters correspond to heavier tails, meaning that the probability of exceeding high intensities decreases in a slower way with increasing intensity, and vice versa. In the study area, the CPM exhibits both overestimation and underestimation, mainly non-significant, of the shape parameter with no evident spatial patterns related to orography (Figure 8b). Indeed, the boxplots show a similar median, just above 1, and similar whiskers for all the elevation groups (Figure 8e). The median bias on scale >1 indicates that in the CPM the distributions generally have lighter tails. Opposite situations occur locally where the shape can be underestimated.

The bias in the average number of yearly ordinary events n is significant in most of the stations (52.3%), and a clear spatial pattern emerges. Strong underestimation is observed in the lowland area and a slight overestimation in the mountainous area (Figure 8c and f). Higher (lower) n translates into higher (lower) estimated return levels.

In terms of orographic relations, the scale parameter in the model increases with elevation and significantly differs from the decreasing scale for observation (Fig. S5a), while the observed relation with elevation for the shape parameter and number of events is better represented by the model (Figure S5b,c,d,f). The CPM overestimation of the return levels in the mountains, and the resulting weaker reverse orographic effect, seem therefore mostly explained by the increasing overestimation of the scale parameter with elevation (Fig. S5b). This indicates a rather homogenous increase of all the ordinary events in the tail, which for the case of hourly durations are the largest 10% of the ordinary events.

### 5.3 A physical-process interpretation of results

In order to ensure no systematic bias was introduced by differences in CPM and rain gauge elevations (that in a few cases is relevant, see Figure 1c), we explore the possible dependence of the magnitude of the bias in the estimated return levels on such differences. Even if we have previously shown that higher biases on return levels are in mountainous areas, these biases (color of dots in Figure S6) are not systematically related to higher elevation differences. We conclude that the elevation difference between SC_CPM and OB could not be considered as the main descriptor of our findings.

Ban et al. (2020) suggested that the CPM overestimation over high elevation areas can be partly related to uncertainty in the observations (gauge undercatch). For the Alpine region, the undercatch of seasonal mean precipitation is found to be about 8% (40%) below 600 m a.sl. (above 1500 m a.sl.) in winter and 4% (12%) in summer (Sevruk, 1985; Richter, 1995). Major possible sources of undercath are related with the tipping movement of the bucket-type rain gauge and with the presence of strong wind. The first tends to affect precipitation intensities that are higher than the ones we observe in our study in the stations at lower elevations; "true" intensities unaffected by undercatch should therefore strengthen our findings about the reverse orographic effect. The latter, depending on the wind-speed, rain gauge shape and precipitation type, could lead to losses of up to 40% for rain and up to 80% for snow at high wind speed (8-10 m/s, Canteruccio et al., 2021). Our study focuses on extreme short-duration rainfall, which is mostly related to convection and is thus less subject to measurement



underestimation of snowfall. In principle, wind-induced undercatch acts irrespective of elevation, but it could be more

relevant in mountainous areas where turbulence and high wind-speeds are more frequent. Part of the CPM overestimation

found at the high elevation could thus be due to this kind of undercatch.

The overestimation of heavy rainfall in high-resolution climate models was also found in previous studies and often linked

with the fact that convection is not fully resolved even at convection-permitting resolutions (Kendon et al. 2021, Ban et.

2020, Panosetti et al. 2020). Indeed, while the grid spacing of our simulation is 2.2 km, the effective resolution is coarser.

Using kinetic energy spectra, Skamarock et al (2004) estimated the effective horizontal resolution of the WRF model (a

model that has a similar dynamical core than COSMO). They found that the shortest horizontal wavelength that is credibly

resolved amounts to typically 5-7 times the grid spacing. Similar results were found in a later study comparing the COSMO

and the ECMWF-IFS model (Zeman et al. 2021). Thus, for our grid spacing, wavelengths smaller than 10-15 km are only

partly resolved. Consistent results were also found in convergence studies. Panosetti et al. (2020) used systematic

convergence experiments with grid spacings in the range of 8 to 0.5 km. They found that structural convergence was not

even achieved at 500 m grid spacing, i.e. the horizontal scale of the convective updrafts narrowed whenever resolution was

refined. However, they found "bulk convergence" in domain-averaged aspects of the flow (such as the probability density

functions of the convective mass flux). In addition, the 2.2 km CPM resolution might not be sufficient to represent fine-scale

orographic features, like the alternation of hills and narrow valleys (see Figure S7), responsible for the development of local

winds and turbulence crucial for triggering convection (Fosser et al. 2015). Moreover, sub-grid processes like shallow

convection, turbulence and microphysics, still use parameterisations formulated for coarser resolution simulations leading to

poor representation of these processes (e.g. Kendon et al. 2021). Marra et al. (2021) also suggested that the observed reverse

orographic effect at short-duration rainfall extremes could be also related to a weakening of the updrafts caused by

orographically-induced turbulence. All the above-mentioned issues could limit the ability of the CPM to fully represent the

interaction of convective cells with orography, thus leading to a bias in the estimation of short duration extremes over this

orographically-complex region. This seems to be confirmed by the significant overestimation of the scale parameter in

mountainous areas, which suggests that short-duration rain rates are almost equally overestimated all along the probability

distribution tail.

Our findings also highlight the complexity of the processes in the lowland and coastal zones, where elevation cannot play a

relevant role. Here, other factors should be considered, such as the distance from the coastline (Marra et al. 2022a) and the

ability of the model to distinguish between sea areas, land areas, and shallow waters (such as the Venice lagoon in our study

case). Further analyses should be carried out to specifically address these issues, for example considering a longer coastline

and additional observational data along the coast and possibly even offshore, for example using weather radars. In

comparison with coarser resolution models (e.g. results in Pichelli et al. 2021) the CPM is known to improve the

representation of hourly extreme rainfall. In the present work CPM estimates are in fact found to provide realistic estimates





of extreme rainfall magnitudes, but the results of the present work show they are not yet suited for providing direct estimation of return levels without proper adjustments.

## 6. Conclusions

In this work, the ability of a km-scale convection-permitting climate model (COSMO-crCLIM at 2.2 km resolution) to
represent extreme short duration precipitation in complex orographic areas is examined. We exploit the potential of a non-asymptotic simplified Metastatistical Extreme Value (SMEV) approach to reduce the stochastic uncertainties related to the use of a short time slice (10 years) to analyse extremes. We focus on the reverse orographic effect, a key feature of extreme precipitation emerging from observational dataset in complex orography. In the eastern Italian Alps, we analysed hourly rainfall data from: 174 rain gauges (our benchmark), 174 station-collocated CPM grid points, and whole grid CPM (~6500
points). We compare 1 h duration annual maxima, return levels up to 100yr, parameters of the SMEV distribution, and we quantified their relation with elevation.

Our findings show that the CPM bias on hourly return levels tends to be positive and tends to increase with elevation. Despite this increasing positive bias with elevation, CPM runs are able to capture the reversed orographic effect, but significantly underestimate its magnitude (~10% of the median per km as opposed to ~30% of the observations). Some
possible explanation for the observed biases may be related to: the "effective resolution" of the CPM model, with a partial representation of convection processes; sub-grid orographically-induced turbulence; insufficiently detailed digital representation of steep valleys in the model; rain gauge undercatch in case of strong wind.

CPMs may be used to investigate high return levels in orographically complex areas poorly covered by observations and to estimate changes in rainfall extremes under future scenarios. However, bias correction approaches need to be developed that
explicitly consider the role of orography (e.g. Velasquez et al, 2020), with specific reference to the case of short-duration extremes. To this end, the potential of non-asymptotic approaches applied on short time slices of CPM simulations could be further explored to improve our understanding of future changes in precipitation extremes. Future works should consider an ensemble of climate models and explore adjustment methods which account for the role of orography at multiple durations.

## Authors contribution

ED: Conceptualization, Data curation, Formal analysis, Methodology, Software, Writing – original draft preparation, Writing – review & editing; FM: Conceptualization, Funding acquisition, Methodology, Software, Supervision, Writing – review & editing; GFos: Conceptualization, Data curation, Funding acquisition, Methodology, Supervision, Writing – review & editing; MM: Conceptualization, Funding acquisition, Supervision, Writing – review & editing; GFor: Data curation,



Writing – review & editing; CS: Data curation, Writing – review & editing; MB: Conceptualization, Funding acquisition, Methodology, Supervision, Writing – review & editing.

**Acknowledgments**

This work was supported by CARIPARO Foundation through the Excellence Grant 2021 to the "Resilience" Project. The authors acknowledge Institutes providing observational data: Provincia Autonoma di Trento, Provincia Autonoma di Bolzano, Agenzia Regionale per la Prevenzione e Protezione Ambientale del Veneto. The authors gratefully acknowledge
the WCRP- CORDEX-FPS on Convective phenomena at high resolution over Europe and the Mediterranean (FPSCONV-ALP-3) and in particular the ETH Zurich for sharing the climate model data.

**Data availability**

The quality-controlled 1h-aggregated rain gauge data used in the study are freely available in the Zenodo repository, at 10.5281/zenodo.7142385 (Dallan, 2022). The CPM data cannot be directly shared by the authors. It should be available
within the WCRP- CORDEX-FPS on Convective phenomena by the end of 2022. The codes used for the statistical model are available at https://doi.org/10.5281/zenodo.3971558 (Marra, 2020).

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
