# Peer review of "How well does a convection-permitting regional climate model represent the reverse orographic effect of extreme hourly precipitation?"

_EGUsphere, 2022_

## Author Comment (AC2)

**Response to referee #2**

The manuscript provides an assessment of whether a high-resolution (2.2 km) regional climate model (COSMO-crCLM) can represent the "reverse orographic effect" - an observed effect whereby short-duration (e.g., hourly) extremes in precipitation decrease with increasing altitude. The study is performed over northeastern Italy including some of the Italian Alps - a region of complex orography. The authors make use of a statistical technique: simplified metastatistical extreme value (SMEV) to reliably determine return times longer than the duration of their simulations (10 yr). They find that the model does produce a reverse orographic effect, albeit with reduced magnitude compared to rain gauge observations.

The manuscript is well written, with clear figures and a logical structure. The interpretation is all backed up by the figures. Uncertainties are assessed using a boostrapping method (random sampling of years with replacement). Potential sources of bias, from underestimation of the rain gauges to differences in the elevations of the gauges and the co-located grid cells, are considered. The discussion further considers why the model might be underestimating the effect, discussing the effective resolution of the model as well as the representation of subgrid processes such as turbulence. All of the points that arose as I was reading the manuscript were addressed in the discussion.

I have therefore recommended publication with technical changes only. Below are some very minor comments and clarifications.

Authors' response: We thank the reviewer for the positive feedback and the provided comments/corrections. We address them in the following. We numbered each comment as R2Cx (Referee 2, Comment x), and our response is indicated with "R" and blue color. In the proposed modifications to the original text, we indicate the new text in *Italics*.

R2C1. General:

- divide symbol (÷) seems to be used where a dash should be used.
- slice/s - I would normally prefer the word period/s
- Fig. vs Figure, consistency.
- I believe collocate should be colocate or co-locate.

R: The suggested corrections will be included in the revised paper.

Specific:

R2C2. Title: suggest changing to "How well does a convection-permitting regional climate model represent the reverse orographic effect of extreme hourly precipitation?"

R: We agree with your suggestion and we will include "regional" in the title.

R2C3. L14: Recent observational studies…

R: We will modify it according to your suggestion.

R2C4. L21: northeastern (no capital)

R: We will modify it according to your suggestion.

R2C5. L45: become -> are becoming

R: We will modify it according to your suggestion.

R2C6. L45: CMPs -> CPMs

R: We will correct the mistake.

R2C7. L77: spell out what the "reverse orographic effect" is: i.e. short-duration extremes decrease with increasing altitude (or similar).

R: We will rephrase adding an explanation of the "reverse orographic effect". It could be something like this: "While the orographic enhancement is also observed for relatively long-duration precipitation extremes (few hours or more), the opposite has been reported for short-duration extremes (hourly and sub-hourly). *This is known as the "reverse orographic effect", i.e. the rainfall intensity decreases with increasing elevation (Avanzi et al., 2015)."*

R2C8. L152: GPU -> GPUs

R: We will modify it according to your suggestion.

R2C9. L153: More details on the physical parameterisations...

R: We will add the following description: "*The model solves numerically the fully compressible governing equations using finite difference methods (Steppeler et al., 2003) on a three-dimensional Arakawa-C grid (Arakawa and Lamb 1977) based on rotated geographical coordinates and a generalized, terrain following height coordinate (Doms and Baldauf 2015). A fifth-order upwind scheme is used for horizontal advection and an implicit Crank-Nicholson scheme in the vertical discretized in 60 stretched model levels ranging from 20 m to 23.5 km (Baldauf et al., 2011). The model employs a third-order Runge-Kutta time-stepping scheme (Wicker and Skamarock, 2002) and a delta-two-stream radiative transfer scheme according to Ritter and Geleyn (1992).The parameterization of precipitation is based on a single-moment bulk cloud microphysics scheme using five categories of hydrometeors, i.e. cloud water, cloud ice, rain, snow, and graupel (Reinhardt and Seifert, 2006). A modified version of the Tiedtke mass flux scheme with moisture convergence closure (Tiedtke, 1989) is used to parameterised shallow convection, while deep convection is resolved explicitly. In the planetary boundary layer and for the surface transfer a turbulent kinetic energy-based parameterization is applied (Mellor and Yamada, 1982; Raschendorfer, 2001) , while in the lower boundary COSMO-crCLIM uses the soil-vegetation-atmosphere-transfer model TERRA-ML with 10-layer soil and a maximum soil depth of 15.24 m (Heise et al., 2006)."*

R2C10. L153-4: Also, it would not hurt to describe some of the key parametrizations here, such as the microphysics (1-moment or 2), and the turbulence parameterization, so that the reader does not have to dig into the references. 2-3 sentences perhaps on these.

R: Please, see the answer above to the previous comment.

R2C11. Fig. 1b: Cannot see OB at low elevations. Perhaps plot as blue line (staircase - i.e. with flat tops/vertical lines) on top of SC_CPM?

R: At low elevation OB and SC_CPM histograms overlap, and the color of the bars is different where they are overlapping (a darker orange). We tried to use a staircase plot, as suggested, but it seems less clear than the histogram. We prefer to keep the histogram, and we will use larger bins (classes of 200 m, as shown below).

[Figure]

[Figure]

R2C12. L218: left-censoring - what does this mean?

R: We will rephrase to: "parameters of the Weibull distribution are calculated by left-censoring the ordinary events below the above-mentioned thresholds *(i.e., censoring their magnitude but retaining their weight in probability)* and using a least-squares linear regression in Weibull transformed coordinates".

R2C13. L244: AM defined as Annual Maxima, i.e. already plural. This means AMs should probably not be used later. (Very pedantic.)

R: We will correct AMs in AM, as suggested.

R2C14. L274: 30% km-1 - please remind reader of the definition without having to refer back to Sect. 3.3.

R: We will add here the definition: "30% km-1 *(expressed as percentage of the median value per km of elevation)* …"

R2C15. L275: Add details of regression $R^2$/fmse?

R: We will add the value of $R^2$ in all the figures showing regression lines (see as example the modified Figure 4).

[Figure]

R2C16. Fig. 2b: suggest adding a vertical dashed line at elevation = 100 m, or showing all points below this as open circles, to visually show they are not included in the regression.

R: Thanks for the suggestion, we will add a vertical dashed line at elevation = 100 m a.s.l. (see modified figure below), and we will update the caption: "[...] slope for the linear regression *(solid line)* is expressed as a percent of the median value and is calculated for the stations above 100 m a.s.l. *(points on the right of the dashed line)*."

[Figure]

R2C17. L284: Figure 3a, b) instead of c)?

R: Yes, thank you for the correction.

R2C18. L286: (b) and (d)

R: Yes, thank you for the correction.

R2C19. L305: missing full stop.

R: Thank you for the correction.

R2C20. Figs. 4, 6: should have a blue triangle in key for SC_CPM (looks green on zooming in to PDF for me)

R: Thank you for this correction, we will update the legend for the SC_CPM with the correct color (blue), in Figure 4 and also Figures 6, S3, S4, S5

R2C21. L348: "The slopes test significantly different at the 5% level." Clarify - slopes of what are different to what?

R: We will modify it as: "The SC_CPM slope is significantly different (5% significance level) from the OB slope".

R2C22. L355: Figure S3 reports the uncertainty in the observed 1 h duration...

R: We will add "observed" as suggested.

L362: ...uncertainty indicate that...

R2C23. R: We will add "indicate" as suggested.

R2C24. L382: it seems to me as if GR_CPM shows less dependence on return period than SC_CPM is worth saying something about. Perhaps one sentence saying this?

R: Thank you for your comment. We will add a sentence about this: "The slopes obtained from the analysis on the whole CPM grid *show a milder decrease for higher return time than the SC_CPM slopes, but since they* are within the uncertainty range of the SC_CPM slopes, *no statistically significant result can be inferred on this.*

R2C25. L392: ...found in a CPM.

R: We will add "a" as suggested.

R2C26. L396: , with the lower values of the interquartile ranges...

R: We will add "lower" as suggested.

R2C27. L398: ...would be double that of the observations...

R: We will correct it.

R2C28. Sect. 5.3: suggest renaming this to "Bias assessment of differences in CPM and rain gauge elevations" (as a short subsection), and moving 5.3 to just before L430.

R: Thank you for the suggestion, but we prefer to keep it as it is, avoiding having an additional very short section.

R2C29. L433: undercath (typo)

R: We will correct: "undercatch".

R2C30. L472: ...estimation of hourly return levels...

R: We will add "hourly" as suggested.

R2C31. L487: ...in the case of strong wind.

R: We will add "the" as suggested.

---

## Author Response (AR1)

**Editor decision**:

Publish subject to revisions (further review by editor and referees) by Nunzio Romano

Comments to the author:

Dear Authors:

Your original submission was evaluated by three reviewers who rated it fairly well and also provided valuable feedback. Most of these comments refer to technical corrections and words to make things clearer and, as required by the reviewers, a quick evaluation of the revised version may be in order.

Therefore, you are invited to submit the revised manuscript allowing for the reviewers' comments and your amendments for improvements. Please, also send a detailed point-by-point reply explaining the revision and any changes you made.

Authors' response: We thank the Editor for the positive feedback and, as requested, we provide here the detailed point-by-point reply to the reviewers. We numbered each comment as RyCx (Referee y, Comment x), and our response is indicated with "R" and blue color. In the modifications to the original text, we indicate the new text in *Italics* and we refer to the line numbering as it is in the revised manuscript.

All the modifications are then included in the uploaded revised manuscript.

**Response to referee #1**

The authors offer a well-presented manuscript examining the ability of a convection-permitting (CP) model (ERA-Interim-driven COSMO-crCLM) to represent the reverse orographic effect at the northeastern Italian Alps area. The manuscript is well-written, and concise, with a good flow and sufficient discussion. The contribution of the manuscript is significant since it gives answers to an issue which may arise for many researchers dealing with CP models.

Every query or suggestion I had during the first part of the manuscript was explained or applied in the next sections, therefore I only have a few minor suggestions, mainly grammatical-syntax comments and typos.

Authors' response: We thank the reviewer for the positive feedback and the provided comments/corrections. We address them in the following. We numbered each comment as R1Cx (Referee 1, Comment x), and our response is indicated with "R" and blue color. In the proposed modifications to the original text, we indicate the new text in *Italics*.

Some minor/discussion comments:

R1C1. It would be helpful to see a short literature review on existing CPM permitting models (probably in the Introduction), and comments on their performance. This would help you justify better the selection of the ERA-Interim-driven COSMO-crCLM.

R: Thanks for the suggestion, we added a short review on CPMs in the introduction, at lines 48-56: "*Thanks to their ability to resolve convective systems and to better represent local processes, CPMs provide more realistic representations of sub-daily precipitation statistics, including the diurnal cycle, spatial structure of precipitation, intensity distribution and extremes (Prein et al. 2015, Berthou et al. 2020, Lind et al., 2016). These added-values have*

*been found using different CPMs over several domains. In additions, CPMs have been proven to better represents temperature especially over mountain regions (e.g. Ban et al., 2014), clouds (e.g. Hentgen et al, 2019), small-scale wind systems (e.g. Belušic et al., 2019), land–atmosphere feedbacks (e.g. Taylor et al, 2013), besides tropical cyclones (e.g. Gentry & Lackmann, 2010) and monsoons (e.g. Marsham et al., 2013). This leads to a greater confidence, especially for short-duration precipitation extremes, in CPM-based projection, compared to coarser resolution models (Kendon et al. 2017, Fosser et al. 2020).”*

In addition, in the description of the convection-permitting model rainfall data (section 2.2) we also added the following clarification (lines 175-180): “*Reanalysis datasets blend in observations and thus provide the best possible lateral boundary conditions to drive a regional model and allow to evaluate the systematic (i.e. not linked to the boundary condition) bias of the model. Ban et al. (2021) evaluated the CPM simulation used here against several observational datasets and found that the bias is limited and comparable within the other CPMs from the Flagship Pilot Study on Convective Phenomena over Europe and the Mediterranean (FPS-Convection; Coppola et al. 2020) run under the Coordinated Regional Climate Downscaling Experiment (CORDEX).*”

R1C2. Lines 17-19: “We introduce the use of a non-asymptotic statistical approach (Simplified Metastatistical Extreme Value, SMEV) for the analysis of extremes from short time slices such as the ones of CPM simulations” The word “introduce” is a bit misleading, since SMEV has already been introduced; maybe rephrase it to “We propose” or something like this?

R: Thanks for this comment. We rephrased accordingly (line 20): “Here, we *use* a non-asymptotic statistical approach [...]”.

R1C3. Lines 55-57: “Over the Alps, but also elsewhere, CPMs tend to generate more precipitation at higher elevations than in reality, thus reducing the bias with respect to observations compared to RCMs (Lind et al. 2016, Reder et al. 2020).” This sentence is confusing to me, it sounds like CPMs overestimate precipitation at higher elevations than in reality, but at the same time, they reduce the bias compared to RCMs. Could you rewrite this?

R: Thank you for pointing this out. The sentence (lines 61-62) have been rephrased to: “Over the Alps, CPMs tend to generate more precipitation at higher elevations *compared to RCMs, thus reducing the bias with observations […]*”.

R1C4. Lines 143-144: “We considered only rain gauges with at least 9 valid years during the period 2000-2009,” Could you explain here why you chose this period?

R: This period corresponds to the CPM one. We added this information at lines 149-150: “*To match the available period in the CPM,* we considered only rain gauges with at least 9 valid years during the period 2000-2009, where a year is defined [...]”

R1C5. Lines 153-154: “More details on the used physical parameterisations can be found in Leutwyler et al. (2016).” Give two-three sentences on the basics of the process.

R: We included the following description (lines 160-172): “*The model solves numerically the fully compressible governing equations using finite difference methods (Steppeler et al., 2003) on a three-dimensional Arakawa-C grid (Arakawa and Lamb 1977) based on rotated geographical coordinates and a generalized, terrain following height coordinate (Doms and*

*Baldauf 2015). A fifth-order upwind scheme is used for horizontal advection and an implicit Crank-Nicholson scheme in the vertical discretized in 60 stretched model levels ranging from 20 m to 23.5 km (Baldauf et al., 2011). The model employs a third-order Runge-Kutta time-stepping scheme (Wicker and Skamarock, 2002) and a delta-two-stream radiative transfer scheme according to Ritter and Geleyn (1992). The parameterization of precipitation is based on a single-moment bulk cloud microphysics scheme using five categories of hydrometeors, i.e. cloud water, cloud ice, rain, snow, and graupel (Reinhardt and Seifert, 2006). A modified version of the Tiedtke mass flux scheme with moisture convergence closure (Tiedtke, 1989) is used to parameterise shallow convection, while deep convection is resolved explicitly. In the planetary boundary layer and for the surface transfer a turbulent kinetic energy-based parameterization is applied (Mellor and Yamada, 1982; Raschendorfer, 2001) , while in the lower boundary COSMO-crCLIM uses the soil-vegetation-atmosphere-transfer model TERRA-ML with 10-layer soil and a maximum soil depth of 15.24 m (Heise et al., 2006)."*

R1C6. Some suggested syntax changes:

R: Thank you for the following suggestions, we included them in the revised manuscript.

Line 25: "SMEV's capability"

Line 26: "promises further applications"

Line 45: "In CMPs,

Lines 51-53: "In areas with a complex terrain, the possibility of explicitly resolving convection along with a more detailed representation of orography and surface properties are crucial elements for correctly capturing the initiation and development of convection"

Line 269: Do you mean "A spatial pattern" instead of "organization"?

Lines 361-363: I think a verb like "show" is missing from that sentence: "The consistency of the return level estimates obtained from the full record and from the 10 yr record, and the small increase in the associated uncertainty show that, once its assumptions are verified, SMEV is a reliable statistical method for the analysis of extreme precipitation from short time slices."

Line 415: "n" in italics

Line 480: "100 yr, and parameters of…"

Figure comments:

R1C7. Figure 2, Figure 4 and rest of the figures showing linear regression:  do you want to also show the coefficient of determination $R^2$?

R: We added the coefficient of determination $R^2$ in all the figures where a linear regression is shown (see example below for modified Figure 4).

[Figure]

R1C8. Figure 4: "(SC_CPM), and all CPM"

R: We added "and" in the caption in Figure 4, and also in Figures 6, S4, S5.

R1C9. Figure 4: "the linear regressions lines shown as a solid line, are expressed as.."

R: We included the suggested modification in the caption of Figure 4, and also in Figures 6, S4, S5.

R1C10. Figure 4: Could you change color for the observations, it is the same as CPM

R: Thank you for pointing this out, we updated the legend for the SC_CPM with the correct color (blue) in Figure 4 and also in Figures 6, S3, S4, S5.

R1C11. Figure 4: You do not focus on the orographic effect for daily but still can show the slope for the 24-hour case

R: The orographic effect at daily duration is rather complicated and doesn't have a unique relation with elevation, as described at lines 77-80: "However, a simple precipitation–height relation is difficult to establish, because the topographic signal is also associated with slope and shielding. In addition, the precipitation increase is robust only for low and intermediate topographic heights. In the Alps, maximum annual mean precipitation is typically in the height range of 800–1200 m (Frei and Schär, 1998), and above this altitude precipitation may again decrease with height." We thus prefer to not show these regressions, but just evaluate the agreement between OB and CPM through the boxplot for different elevation classes.

R1C12. Figure 7: remove "," from: "grid, CPM"

R: We corrected it.

R1C13. Figure 7: "are significant" instead of "result significant;"

R: We modified it.

**Response to referee #2**

The manuscript provides an assessment of whether a high-resolution (2.2 km) regional climate model (COSMO-crCLM) can represent the "reverse orographic effect" - an observed effect whereby short-duration (e.g., hourly) extremes in precipitation decrease with increasing altitude. The study is performed over northeastern Italy including some of the Italian Alps - a region of complex orography. The authors make use of a statistical technique: simplified metastatistical extreme value (SMEV) to reliably determine return times longer than the duration of their simulations (10 yr). They find that the model does produce a reverse orographic effect, albeit with reduced magnitude compared to rain gauge observations.

The manuscript is well written, with clear figures and a logical structure. The interpretation is all backed up by the figures. Uncertainties are assessed using a boostrapping method (random sampling of years with replacement). Potential sources of bias, from underestimation of the rain gauges to differences in the elevations of the gauges and the co-located grid cells, are considered. The discussion further considers why the model might be underestimating the effect, discussing the effective resolution of the model as well as the representation of subgrid processes such as turbulence. All of the points that arose as I was reading the manuscript were addressed in the discussion.

I have therefore recommended publication with technical changes only. Below are some very minor comments and clarifications.

Authors' response: We thank the reviewer for the positive feedback and the provided comments/corrections. We addressed them in the following. We numbered each comment as R2Cx (Referee 2, Comment x), and our response is indicated with "R" and blue color. In the modifications to the original text, we indicate the new text in *Italics*.

R2C1. General:

- divide symbol (÷) seems to be used where a dash should be used.
- slice/s - I would normally prefer the word period/s
- Fig. vs Figure, consistency.
- I believe collocate should be colocate or co-locate.

R: The suggested corrections are included in the revised paper.

Specific:

R2C2. Title: suggest changing to "How well does a convection-permitting regional climate model represent the reverse orographic effect of extreme hourly precipitation?"

R: We agree with your suggestion and we included "regional" in the title.

R2C3. L14: Recent observational studies…

R: We modified it (line 16) according to the suggestion.

R2C4. L21: northeastern (no capital)

R: We modified it (line 23) according to your suggestion.

R2C5. L45: become -> are becoming

R: We modified it (line 47) according to your suggestion.

R2C6. L45: CMPs -> CPMs

R: We corrected the mistake (line 47).

R2C7. L77: spell out what the "reverse orographic effect" is: i.e. short-duration extremes decrease with increasing altitude (or similar).

R: We rephrased adding an explanation of the "reverse orographic effect" (lines 80-83): "While the orographic enhancement is also observed for relatively long-duration precipitation extremes (few hours or more), the opposite has been reported for short-duration extremes (hourly and sub-hourly). This is known as the "reverse orographic effect", i.e. the rainfall intensity decreases with increasing elevation (Avanzi et al., 2015)."

R2C8. L152: GPU -> GPUs

R: We modified it (line 159) according to your suggestion.

R2C9. L153: More details on the physical parameterisations...

R: We added the following description at section 2.2 (lines 160 -172): "The model solves numerically the fully compressible governing equations using finite difference methods (Steppeler et al., 2003) on a three-dimensional Arakawa-C grid (Arakawa and Lamb 1977) based on rotated geographical coordinates and a generalized, terrain following height coordinate (Doms and Baldauf 2015). A fifth-order upwind scheme is used for horizontal advection and an implicit Crank-Nicholson scheme in the vertical discretized in 60 stretched model levels ranging from 20 m to 23.5 km (Baldauf et al., 2011). The model employs a third-order Runge-Kutta time-stepping scheme (Wicker and Skamarock, 2002) and a delta-two-stream radiative transfer scheme according to Ritter and Geleyn (1992). The parameterization of precipitation is based on a single-moment bulk cloud microphysics scheme using five categories of hydrometeors, i.e. cloud water, cloud ice, rain, snow, and graupel (Reinhardt and Seifert, 2006). A modified version of the Tiedtke mass flux scheme with moisture convergence closure (Tiedtke, 1989) is used to parameterise shallow convection, while deep convection is resolved explicitly. In the planetary boundary layer and for the surface transfer a turbulent kinetic energy-based parameterization is applied (Mellor and Yamada, 1982; Raschendorfer, 2001), while in the lower boundary COSMO-crCLIM uses the soil-vegetation-atmosphere-transfer model TERRA-ML with 10-layer soil and a maximum soil depth of 15.24 m (Heise et al., 2006)."

R2C10. L153-4: Also, it would not hurt to describe some of the key parametrizations here, such as the microphysics (1-moment or 2), and the turbulence parameterization, so that the reader does not have to dig into the references. 2-3 sentences perhaps on these.

R: Please, see the answer above to the previous comment.

R2C11. Fig. 1b: Cannot see OB at low elevations. Perhaps plot as blue line (staircase - i.e. with flat tops/vertical lines) on top of SC_CPM?

R: At low elevation OB and SC_CPM histograms overlap, and the color of the bars is different where they are overlapping (a darker orange). We tried to use a staircase plot, as suggested, but it seemed less clear than the histogram. We prefer to keep the histogram, and we modified figure 1b by using larger bins (classes of 200 m, as shown below, instead of 100 m as in the previous version).

[Figure]

R2C12. L218: left-censoring - what does this mean?

R: We rephrased to (lines 242-244): "[…] parameters of the Weibull distribution are calculated by left-censoring the ordinary events below the above-mentioned thresholds *(i.e., censoring their magnitude but retaining their weight in probability)* and using a least-squares linear regression in Weibull transformed coordinates".

R2C13. L244: AM defined as Annual Maxima, i.e. already plural. This means AMs should probably not be used later. (Very pedantic.)

R: We corrected AMs in AM, as suggested.

R2C14. L274: 30% km-1 - please remind reader of the definition without having to refer back to Sect. 3.3.

R: We added at line 299 the definition: "30% km$^{-1}$ *(expressed as percentage of the median value per km of elevation, and* computed using the rain gauges above 100 m a.s.l.), …".

R2C15. L275: Add details of regression R^2/fmse?

R: We added the value of $R^2$ in all the figures showing regression lines (see as example the modified Figure 4).

[Figure]

R2C16. Fig. 2b: suggest adding a vertical dashed line at elevation = 100 m, or showing all points below this as open circles, to visually show they are not included in the regression.

R: Thanks for the suggestion, we added a vertical dashed line at elevation = 100 m a.s.l. (see modified figure below), and we updated the caption: "[...] slope for the linear regression *(solid line)* is expressed as a percent of the median value and is calculated for the stations above 100 m a.s.l. *(points on the right of the dashed line)*."

[Figure]

R2C17. L284: Figure 3a, b) instead of c)?

R:  Yes, thank you for the correction (line 310).

R2C18. L286: (b) and (d)

R: Yes, thank you for the correction (line 312).

R2C19. L305: missing full stop.

R: Thank you for the correction (line 330).

R2C20. Figs. 4, 6: should have a blue triangle in key for SC_CPM (looks green on zooming in to PDF for me)

R: Thank you for this correction, we updated the legend for the SC_CPM with the correct color (blue), in Figure 4 and also Figures 6, S3, S4, S5

R2C21. L348: "The slopes test significantly different at the 5% level." Clarify - slopes of what are different to what?

R: We modified it (line 375): "The SC_CPM slope is significantly different (5% significance level) from the OB slope".

R2C22. L355: Figure S3 reports the uncertainty in the observed 1 h duration...

R: We added "observed" as suggested (line 383).

L362: ...uncertainty indicate that...

R2C23. R: We added "indicate" as suggested (line 390).

R2C24. L382: it seems to me as if GR_CPM shows less dependence on return period than SC_CPM is worth saying something about. Perhaps one sentence saying this?

R: Thank you for your comment. We added a sentence about this (lines 408-410): "The slopes obtained from the analysis on the whole CPM grid *show a milder decrease for higher return time than the SC_CPM slopes, but since they* are within the uncertainty range of the SC_CPM slopes, *no statistically significant result can be inferred on this.*

R2C25. L392: ...found in a CPM.

R: We added "a" as suggested (line 420).

R2C26. L396: , with the lower values of the interquartile ranges...

R: We added "lower" as suggested (line 424).

R2C27. L398: ...would be double that of the observations...

R: We corrected it (line 426).

R2C28. Sect. 5.3: suggest renaming this to "Bias assessment of differences in CPM and rain gauge elevations" (as a short subsection), and moving 5.3 to just before L430.

R: Thank you for the suggestion, but we prefer to keep it as a paragraph in the current section, avoiding having an additional very short section. This section has been numbered as 5, since previous subsections 5.1 and 5.2 have been moved under the results section, as suggested by referee 3.

R2C29. L433: undercath (typo)

R: We corrected in "undercatch".

R2C30. L472: ...estimation of hourly return levels...

R: We added "hourly" as suggested (line 500).

R2C31. L487: ...in the case of strong wind.

R: We added "the" as suggested (line 515).

**Response to referee #3**

The manuscript assesses the representation of the extreme precipitation by the convection-permitting-scale dynamically downscaled regional climate model COSMO-CLM, driven by ERA-Interim reanalysis. The focus of the study lies on the reproducibility of the "reverse orographic effect" consisting of a decrease of short-duration extreme precipitation with the increase of the elevation in the complex-orography context of northeastern Italy. To limit drawbacks in terms of underrepresented climate variability within the short temporal segment of 10 years considered and related large uncertainty on the estimation return period longer than the available period, Authors take advantage of the Simplified Metastatistical Extreme Value (SMEV). This approach relies on the assumption that a suitable statistical model describing the ordinary events may be identified and related distribution can be used to define the distribution of yearly maxima and to capture the probability of occurrence of extremes. Uncertainties are characterized using a bootstrapping method and sources of bias from rain gauges located at different elevations have been taken into account.

The manuscript is overall well written, and results regarding the adoption of the statistical SMEV approach and physical mechanisms behind the presented results (i.e., subgrid processes behind the model underestimation of the reverse orographic effect) have been comprehensively presented and properly discussed.

Authors' response: We thank the reviewer for the positive feedback and the provided comments/corrections. We address them in the following. We numbered each comment as R3Cx (Referee 3, Comment x), and our response is indicated with "R" and blue color. In the modifications to the original text, we indicate the new text in *Italics*.

It follows only minor comments.

R3C1. Line 45: CPMs instead of CMPs.

R: We corrected the mistake (line 47).

R3C2. Line 60: Please correct the doubled reference.

R: The references Poschlod et al. 2021 and Poschlod 2021 (line 66) refer to two different papers.

R3C3. Lines 205-207: This statement is not supported by evidence. Please provide a real demonstration through some plots in defense of the applicability of Weibull tail approximation to the right distribution tail of ordinary events of your datasets.

R: We include below here an example of how the Weibull tails fit the observed data, but we need to point out that we disagree with this evaluation. We tested whether the H0 hypothesis of having Weibull tails can be rejected by the available observations. To do so, we used long rain gauge records and we checked whether the observed annual maxima could be likely samples from Weibull tails estimated explicitly censoring the observed annual maxima themselves (censoring their magnitude and retaining their weight in probability). Such an approach provides outcomes that are more robust than a visual evaluation of the goodness of fit.

[Figure]

The figure shows an example of Weibull fit of the tail of ordinary events for one station, two different durations (1h and 24h), and a left-censoring threshold of 0.75. Points represent all the ordinary events, orange points represent the events used for the Weibull fit (here the top 25%). The dashed lines indicate the Weibull distribution fitting the data, and the shaded areas indicate the 90% Weibull sampling uncertainty, obtained with a bootstrap procedure. On the x-axis, p represents the non-exceedance probability. If the portion of annual maxima outside the shaded area is more than 10%, the hypothesis of Weibull tail is rejected, for the tested left-censoring threshold.

R3C4. Lines 214-215: Please better introduce this section.

R: We modified the introduction to the section 3.1.2 as follow: "*Extreme return levels are estimated using* the SMEV statistical model  as described in Marra et al. (2020) [...]"

R3C5. Line 252: Correct the numbering of the section.

R: Thank you for pointing it out, we corrected the numbering in 3.4.

R3C6. Line 348: "The slopes test significantly different at the 5% level." Not clear what "different" is referring to.

R: We modified line 375 as: "The SC_CPM slope is significantly different (5% significance level) from the OB slope".

R3C7. Why place subsections 5.1 and 5.2 in the discussion section and not in the results section?

R: Thank you for this suggestion, we have moved the two subsections in the results section, and numbered as 4.4 and 4.5 respectively.

R3C8. Figures 4 and 6 it quite challenging distinguishing green and blue markers.

R: In the legend, we corrected the color for SC_CPM (it was green by mistake).